# Deregulated miRNA Expression in Triple-Negative Breast Cancer of Ancestral Genomic-Characterized Latina Patients

**DOI:** 10.3390/ijms241713046

**Published:** 2023-08-22

**Authors:** Maram Almohaywi, Bruna M. Sugita, Ariana Centa, Aline S. Fonseca, Valquiria C. Antunes, Paolo Fadda, Ciaran M. Mannion, Tomilowo Abijo, Stuart L. Goldberg, Michael C. Campbell, Robert L. Copeland, Yasmine Kanaan, Luciane R. Cavalli

**Affiliations:** 1Microbiology Department, Howard University Cancer Center, Howard University, Washington, DC 20059, USA; 2Research Institute Pelé Pequeno Príncipe, Faculdades Pequeno Príncipe, Curitiba 80250-060, PR, Brazil; 3Genomics Shared Resource, Comprehensive Cancer Center, The Ohio State University, Columbus, OH 43210, USA; 4Department of Pathology, Hackensack University Medical Center, Hackensack, NJ 07701, USA; 5National Institute of Diabetes and Kidney Diseases, National Institute of Health, Bethesda, MD 20814, USA; 6John Theurer Cancer Center, Hackensack Meridian School of Medicine, Hackensack, NJ 07701, USA; 7COTA, Inc., New York, NY 10014, USA; 8Department of Biological Sciences Human and Evolutionary Biology Section, University of Southern California, Los Angeles, CA 90089, USA; 9Pharmacology Department, Howard University Cancer Center, Howard University, Washington, DC 20059, USA; 10Oncology Department, Lombardi Comprehensive Cancer Center, Georgetown University, Washington, DC 20007, USA

**Keywords:** triple-negative breast cancer, Latinas, copy number alterations, microRNA, miRNA

## Abstract

Among patients with triple-negative breast cancer (TNBC), several studies have suggested that deregulated microRNA (miRNA) expression may be associated with a more aggressive phenotype. Although tumor molecular signatures may be race- and/or ethnicity-specific, there is limited information on the molecular profiles in women with TNBC of Hispanic and Latin American ancestry. We simultaneously profiled TNBC biopsies for the genome-wide copy number and miRNA global expression from 28 Latina women and identified a panel of 28 miRNAs associated with copy number alterations (CNAs). Four selected miRNAs (miR-141-3p, miR-150-5p, miR-182-5p, and miR-661) were validated in a subset of tumor and adjacent non-tumor tissue samples, with miR-182-5p being the most discriminatory among tissue groups (AUC value > 0.8). MiR-141-3p up-regulation was associated with increased cancer recurrence; miR-661 down-regulation with larger tumor size; and down-regulation of miR-150-5p with larger tumor size, high p53 expression, increased cancer recurrence, presence of distant metastasis, and deceased status. This study reinforces the importance of integration analysis of CNAs and miRNAs in TNBC, allowing for the identification of interactions among molecular mechanisms. Additionally, this study emphasizes the significance of considering the patients ancestral background when examining TNBC, as it can influence the relationship between intrinsic tumor molecular characteristics and clinical manifestations of the disease.

## 1. Introduction

Breast cancer occurrence differs significantly across women from various races and ethnicities. In 2023, it was projected that there will be 297,790 new cases of invasive breast cancer and 43,170 deaths in the United States (USA) (https://seer.cancer.gov/statfacts/html/breast.html, accessed on 7 August 2023). Among Hispanic/Latina (herein called Latina) women, who make up 18.9% of the USA population (www.census.gov), an estimated 28,100 breast cancer cases and 3100 fatalities were anticipated in 2021 (ACS, Cancer Facts & Figures for Hispanic/Latino People 2021–2023).

The distribution of breast cancer subtypes also exhibits notable variation based on race and ethnicity [1,2,3,4]. The triple-negative breast cancer (TNBC) subtype—negative for estrogen, progesterone, and HER2/NEU receptors, for instance—is more commonly diagnosed in women of African descent and Latinas compared to Non-Hispanic Whites (NHW) [2,5,6]. Additionally, these groups exhibit distinct disease presentations upon diagnosis. African American (AA) women are more likely to develop breast cancer at younger ages, with more advanced stages, and with less localized disease when compared to other groups [5,6,7,8]. These differences may be attributed to various well-known socioeconomic, cultural, and biological factors that directly influence treatment response and survival rates [6,7,9,10].

The TNBC subtype encompasses a diverse range of subtypes distinguished by unique molecular profiles, which can lead to varying responses to therapy and prognoses [11,12,13,14]. The utilization of multi-omics approaches to further refine these molecular profiles has greatly enhanced prognostic accuracy and has played a pivotal role in the advancement of targeted therapies, particularly those centered around immune checkpoint inhibitors [15,16,17,18]. Nevertheless, there is a lack of comprehensive research on the molecular profiles of TNBC in specific ethnic populations, such as Latinas [19,20]. As a result, there is a gap in understanding the biological characteristics of their tumor patterns and, consequently, in identifying relevant biomarkers and their correlation with clinical presentation, treatment options, and prognosis.

MicroRNAs (miRNAs) are small non-coding regulators of the expression of genes involved in key biological processes. MiRNA expression have been shown to vary among different ancestries. Several mechanisms can contribute to this variability, including genetic variations, such as single nucleotide polymorphisms (SNPs) and other genetic changes, that can influence miRNA biogenesis and function, leading to differences in miRNA expression between populations [21,22,23]. In addition, natural selection and environmental factors may contribute to the enrichment or depletion of specific miRNA variants in different ancestries, affecting the miRNA regulation of genes involved in disease pathways [24,25]. Epigenetic modifications and gene–environment interactions can further contribute to the diversity in miRNA expression and function across populations [26]. These mechanisms can confer variations and distinctly impact the susceptibility and prevalence to certain diseases, treatment responses, and outcomes. Understanding these mechanisms is essential for comprehending disease susceptibility and tailoring personalized therapies for different ancestral groups.

In cancer, dysregulation of the miRNAs profile has been associated with sustaining cell proliferation, activating invasion, and supporting metastasis [27,28]. In breast cancer, gene expression profiling has identified distinct patterns of miRNA expression among various tumor subtypes [29,30,31,32]. In TNBC, several studies have reported deregulated miRNA expression in association with more aggressive phenotypes [33,34,35,36,37,38,39].

MiRNA profiling of the TNBC tumor cells (and of several other types of tumors) can vary according to the populational group studied [20,26,39,40,41,42,43,44]. Most of the data are, however, in the comparison of NHW to AA TNBC patients [26,40,44,45,46,47]. In our previous study, we reported that ancestrally characterized AA patients, when compared to NHW, exhibited distinct patterns of copy number alterations (CNAs) and miRNA expression profiles in both their TNBC and non-TNBC tumor tissues [44]. Similarly, an investigation of Latina patients with TNBC residing in Brazil revealed a unique pattern of alterations affecting signaling pathways and associated with poor prognosis [20]. Collectively, these studies indicate that the existence of intrinsic molecular signatures of TNBC that are specific to racial and/or ethnic groups can significantly impact the patients ’clinical outcome.

In this study, our primary aim was to comprehensively characterize the molecular signature of TNBC in a specific cohort of ancestrally defined Latina patients to determine the impact on prognosis and potential clinical outcomes. We conducted a simultaneous analysis of the genome-wide copy number and global miRNA expression in TNBC samples from ancestral Latina patients living in New Jersey, USA. The integrated molecular TNBC signatures, performed in the same tissue samples, were compared against clinical–pathological data obtained from de-identified (coded) patient charts, encompassing disease presentation, patient comorbidities, treatments, tumor stage, presence of distant metastases, cancer recurrence rates, and survival outcomes. In addition, four of the identified miRNAs (miR-141-3p, miR-150-5p, miR-182-5p, and miR-661) with relevance to the TNBC phenotype were individually validated by RT-qPCR and associated with the clinical–pathological data.

## 2. Results

### 2.1. Copy Number Alterations (CNAs) Analysis

CNAs analysis was performed by array comparative genomic hybridization (array-CGH) in 85.7% (24/28) of the formalin-fixed, paraffin-embedded (FFPE) TNBC cases. Two-hundred and twelve (212) CNAs (as measured by “number of calls”) were identified in the cases, with an average of 8.8 ± 11.8 CNAs per case. The combined copy number profile of the analyzed cases is presented in Figure 1. The most frequently affected cytobands were 1q21.1–q24.2 (29%), 3q26.1–q27.2 (23.8%), 4p16.3–p15.31 (19%), 5q21.1–q35.3 (19%), 6p25.3–p24.2 (33.3%), 6p22.3–p21.32 (19%), 8q13–q24.3 (33.3%), 8q24.3 (33.3%), 11q13.2–q13.3 (19%), 19p13.3–p13.11 (33.33%), 21q21.3–q22.3 (19%), Xp22.33 (33.3% with start site 1179089 and stop 2353577), Xp22.33 (23.8% with start site 218292 and stop 2622294), and Xp22.33–p22.2 (47.6%) (Table 1)**.** A total of 2584 genes and 226 miRNAs were mapped on the selected cytoband affected with copy number gains, and 522 genes and 72 with copy number losses (Appendix A).

### 2.2. Global miRNA Expression profiling

MiRNA expression profiling was performed in 64.3% (18/28) of the FFPE TNBC cases. Three hundred and eighty-one (381) miRNAs were found differentially expressed (DE) between the TNBC cases and the non-TNBC controls (32 hormone-positive breast cancer samples) (*t*-test *p* < 0.01, FDR < 0.05) (Figure 2). Most of the TNBC cases were observed clustered, except for two cases (cases # 4 and 6). The top 15 most DE miRNAs (up-regulated and down-regulated) between the TNBC and non-TNBC subtypes are presented in Table 2. The complete list of DE miRNAs is presented in Appendix A.

### 2.3. Integration of miRNA Expression and Copy Number Alterations (CNAs) Analysis

To determine whether CNAs could be one of the mechanisms that lead to alterations in miRNA expression, we performed a direct integration of the array-CGH and miRNA profiling data. Eighteen cases that were profiled for miRNA expression were also analyzed for CNAs. The first integration, which consisted of the mapping of the DE miRNAs in the cytobands most affected by CNAs, revealed 28 miRNAs (28/381 = 7.4%) that were mapped on these cytobands (Table 3). Sixteen of the miRNAs (16/28 = 57%) presented expression alterations on same direction of copy number: thirteen miRNAs were up-regulated in the TNBC and mapped in cytobands with copy number gains (miR-1204, miR-1224-5p, miR-1236-3p, miR-2053, miR-3150b-3p, miR-3151-5p, miR-4448, miR-548d-3p, miR-548d-5p, miR-638, miR-661, miR-6721-5p, and miR-765), and three miRNAs were down-regulated in the TNBC cases and mapped in cytobands with copy number losses (miR-145-5p, miR-146a-5p, and miR-218-5p)**.**

Kyoto Encyclopedia of Genes and Genomes (KEGG) pathways analysis of the 16 DE miRNAs mapped on CNAs resulted in 56 pathways potentially affected by these miRNAs. The ones affected by the largest number of miRNAs (15/16) included adherent junction and focal adhesion; proteoglycans in cancer; pathways in cancer; cAMP, Rap1, and Ras signaling pathways; and signaling pathways regulating pluripotency of stem cells (Appendix A).

Our second integration analysis aimed to determine if any genes might be affected by both mechanisms: copy number and miRNA expression deregulation. A list of 9814 genes was predicted to be targeted by the selected 16 DE miRNAs (each gene was predicted to be targeted by one to ten miRNAs). A comparison between this list and the genes observed in CNAs resulted in 867 genes, among them the *ZNF704* (targeted by 10 miRNAs), *MMP16* (targeted by eight miRNAs), and the *KCNN3, POU2F1, ADARB1, MPZL1, RUNX1T1, UBE2W,* and *DYRK1A* genes (each targeted by seven miRNAs) (Appendix A).

### 2.4. The Cancer Genomic Atlas (TCGA) miRNA Analysis

To further determine whether the above-observed DE miRNAs (381 miRNAs, including the ones that were present in regions with CNAs) were also DE in other breast cancer cases from Hispanic/Latina populations, a search in The Cancer Genome Atlas Breast Invasive Carcinoma (TCGA-BRCA) database was performed. From the limited number of BRCA cases available with the reported Hispanic/Latina ethnicity information (TNBC: n = 7, non-TNBC: n = 26), 99 DE miRNAs were observed between the TNBC and non-TNBC cases (*t*-test *p* < 0.05) (Appendix A).

A comparison of the DE miRNAs of the TCGA analysis with the DE miRNAs of the global analysis of our cases resulted in 33 miRNAs in common, with 23 of them with the same miRNA expression direction: 21 miRNAs were found down-regulated (let-7a-5p, let-7b-5p, let-7f-5p, let-7g-5p, miR-10a-5p, miR-10b-5p, miR-181c-5p, miR-191-5p, miR-195-5p, miR-200a-3p, miR-200b-3p, miR-26a-5p, miR-26b-5p, miR-29b-3p, miR-30a-5p, miR-30b-5p, miR-342-3p, miR-34a-5p, miR-423-5p, and miR-664a-3p), and two up-regulated (miR-146b-3p and miR-766-3p) (Table 4).

### 2.5. Selection of the DE miRNA and miRNA-mRNA Network

To validate the individual differential expression of the miRNAs identified in our cases’ global miRNA profiling and the TCGA dataset, we searched for the experimentally validated interactions of the miRNAs and their target genes that present relevance to breast cancer. By using the Integrated Breast Cancer Pathway (Wikipathways), we selected four miRNAs for validation based on the highest log2FC: miR-141-3p (with down-regulated expression in our Latina samples), miR-150-5p (with down-regulated expression in both analyses: our samples and TCGA), miR-182-5p (with deregulated expression in the TCGA cases), and miR-661 (with up-regulated expression in our samples). A network with experimentally validated target genes of the selected four miRNAs that were involved in the Integrated Breast Cancer Pathway was constructed. Among the several miRNA–mRNA target interactions in TNBC and/or breast cancer in general, the cancer driver genes, such as *BRCA1*, *ESR1*, *PTEN*, and *AKT1,* were observed. MiR-141-3p and miR-182-5p were the miRNAs with the most central interactions that directly regulate a higher number of gene targets (Figure 3).

### 2.6. Validation of the Selected DE miRNA

To validate the individual expression level of each of the above-selected miRNAs, as well as to determine its specificity to the tumor cells, RT-qPCR was performed in 22 tumor tissues (78.6% of the cases (22/28)) and 12 corresponding adjacent non-tumor (ANT) tissue (54% of the cases (12/22)) of the TNBC Latina cases of our study.

For each of the four miRNAs, the expression was first determined between the two groups of breast tissues (tumor vs. ANT). This analysis showed significant DE of miR-150-5p (down-regulated, unpaired *t*-test, *p* < 0.05) and miR-182-5p (up-regulated, unpaired *t*-test, *p* ≤ 0.05) in the tumor when compared to the ANT tissue groups. For the miR-141-3p and miR-661, although not significant (unpaired *t*-test, *p* = 0.08 and *p* = 0.06, respectively), there was a trend for the up-regulation of miR-141-3p and the down-regulation of miR-661 in the tumor when compared to the ANT tissue groups (Figure 4A). However, when this analysis was conducted only in the subset of the paired cases (tumor and ANT tissues) from the same patient (n = 12) was the miR-141-3p observed as significantly DE between the tumor and ANT tissues (unpaired *t*-test, *p* = 0.0194). MiR-661 remained with no significant expression difference (*p* = 0.07). Next, the expression levels of the miRNAs were evaluated between each of the 12 paired cases of tumor and ANT tissue. This analysis showed significant results: miR-141-3p and miR-150-5p were DE in the tumor and ANT in seven cases, and miR-182-5p and miR-661 in eight cases. Variable levels of expression were observed for miR-141-3p and miR-150-5p among the paired cases, whereas only down-regulated expression was observed for miR-182-5p and miR-661 (Figure 4B).

### 2.7. Discriminatory Power of the Selected DE miRNA

The expression levels of the four selected miRNAs were evaluated for their power in discriminating the TNBC and ANT tissues of the patients. Receiver operating characteristic (ROC) analysis showed that 75% of the miRNAs presented an area under the curve (AUC) value superior to 0.7. This analysis was performed for all the TNBC vs. ANT tissues (Figure 5A) and the paired tumor and ANT tissue samples (Figure 5B). MiR-182-5p was the one that presented the highest discriminatory power, with AUC values ≥ 0.8. The combined analysis of the four miRNAs showed an AUC value of 0.5051 for all TNBC vs. ANT tissue samples and an AUC value of 0.6143 for the matched tumor and ANT tissues. As for the combination of the four miRNAs, when the analysis was performed by pairs or trios of the miRNAs, the AUC value was not significant, demonstrating a higher individual discriminatory power of miR-182-5p compared to any combination of the four studied miRNAs (Appendix A).

### 2.8. Association of miR-141-5p, miR-150-5p, miR-182-3p, and miR-661 Expression with the Clinical Parameters of the TNBC Latina Patients

The levels of expression of the four miRNAs obtained by RT-qPCR in the above analyzed TNBC cases were associated with clinical–pathological parameters of the patients (mean age at diagnosis, tumor size, grade and stage, expression levels of ki67 and p53), patients’ co-morbidities and mean body mass index (BMI) values, as well as with follow-up data (breast cancer recurrence, distant metastasis, and survival status). The number of patients analyzed for each of these variables varied for each analyzed miRNA (Table 5). MiR-150-5p presented the highest number of associations with the analyzed parameters; its down-regulation was associated with larger tumor size, the higher expression level of p53 protein, increased breast cancer recurrence, presence of distant metastasis, and deceased status. MiR-141-3p up-regulation was associated with breast cancer recurrence, and miR-661 down-regulation was associated with tumor size, whereas no association of miR-182-5p with any of the analyzed parameters was observed. A multivariate analysis was performed, and, except for miR-661 which was significantly associated with tumor grade (*p* = 0.028), there were no significant associations between the miRNAs’ expression and the clinical variables (Appendix A).

**Figure 5 ijms-24-13046-f005:**
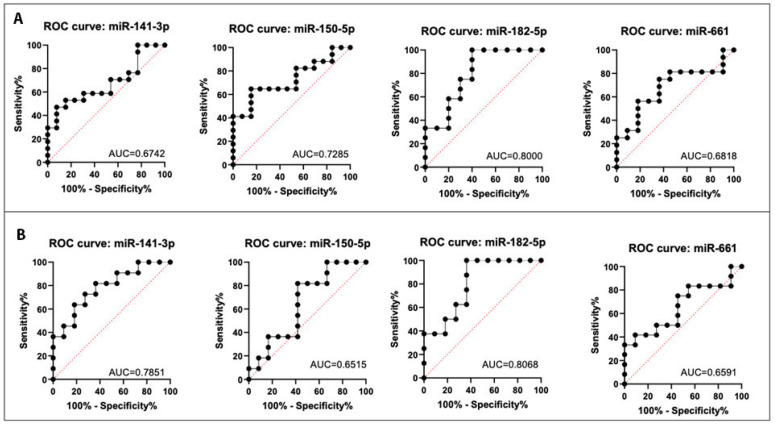
ROC analysis of the non-paired (**A**) and paired (**B**) tumor and adjacent non-tumor (ANT) tissues for the miR-141-3p, miR-150-5p, miR-181a-5p, and miR-182-5p.

### 2.9. Survival Analysis

The different expression of the miRNAs (low or high expression) was evaluated in respects of patients’ survival. No significant associations were observed in both single and paired miRNA analysis. We then verified the association of miRNA expression and survival in the breast cancer cases of the Molecular Taxonomy of Breast Cancer International Consortium (METABRIC) dataset. A higher expression of miR-141-3p was associated with a short survival (Figure 6A) of the breast cancer patients in general (*p* <0.0001). In the TNBC breast cancer patients, however, the association between the expression level of this miRNA and survival was not significant (Figure 6B). Low expression of miR-150-5p was associated with worse survival for breast cancer patients of all subtypes (Figure 6C), and for TNBC patients only (Figure 6D), with values of *p* <0.01 and *p* <0.001, respectively. MiR-182-5p expression was not significantly associated with survival in any of the breast cancer patients’ groups, and miR-661 expression was not found for analysis in this database. On the other hand, when analyzing the available breast cancer patients’ data in the TCGA database, the expression of miR-141-3p was not significantly associated with survival. Also, contrary to the METABRIC dataset, low expression of miR-150-5p was significantly associated with worse survival in all breast cancer cases (*p* < 0.001) but was not significant for the TNBC cases. Low miR-182-5p expression was associated with lower survival when evaluating all breast cancer cases (*p* < 0.05) (Figure 6E) but appeared to improve survival in TNBC cases (*p* < 0.05) (Figure 6F). Low expression of miR-661 was associated with higher survival when evaluating all cases (*p* < 0.001) (Figure 6G) and worse survival for the TNBC cases only (*p* < 0.01) (Figure 6H).

## 3. Discussion

In breast cancer, the classification of breast tumor intrinsic subtypes relies on well-defined and established genome-wide molecular signatures [13,48,49,50,51]. However, prior studies have often overlooked patient ancestry, which can lead to potential inaccuracies in representing the molecular signatures of the distinct ancestral groups within the subtypes. Moreover, by not accounting for ancestry, these studies do not illustrate the extent of molecular variability that exists among and within these groups [52,53,54], particularly in populations with diverse ancestral backgrounds such as Latinas.

There is limited knowledge regarding the genomic signatures of TNBC in Latinas [55,56,57,58,59]. The Cancer of the Genome Atlas (TCGA) database, which includes multi-omics genome-wide profiling data, lists only 34 (out of 770) breast cancer cases from patients of Hispanic and/or Latin American ancestry (21 cases of luminal A subtype, five of luminal B, two of HER2, and six of basal) (https://www.cancer.gov/tcga, accessed 23 March 2023). The ancestral analysis conducted in our study, initially selected based on self-reported race/ethnic information, revealed that the patients had genetic roots in Latin America, specifically in countries such as Peru, Mexico, Colombia, and Puerto Rico. Our subjects did not have European or African backgrounds.

Copy number alterations indicate genomic patterns within specific chromosome cytobands that present as gains/amplifications and/or losses/deletions of genetic material [60,61]. In our array-CGH analysis of 24 TNBC tissue samples, we identified the following cytobands commonly affected: 1q21.1-q24.2, 6p25.3-p24.2, 8q13-q24.3, 8q24.3, 19p13.3-p13.11, Xp22.33, and Xp22.3-p22.2. Some of these cytobands (e.g., 1q21.1, 6p25.3, 8q21.3-q24.3, and Xp22.33) were the same as those that we previously reported among Brazilian patients with TNBC [20]. However, the CNAs on 1q21.1, 6p25.3, 8q11.1-q24.3, 19p13.3, and Xp22.3 cytobands were also described as altered in South African and AA patients with TNBC [44,62], and may therefore not be race-/ethnic-specific CNAs.

In the miRNA profiling analysis of the TNBC cases of this study, 381 miRNAs were differentially expressed compared to controls (non-TNBC cases). Our analysis correctly classified sample subtypes, except for two TNBC cases (samples # 4 and 6), by independent clustering of TNBC and non-TNBC cases.

Taking into account that miRNAs have been observed to be preferentially located in regions of genomic instability, which are characterized by the presence of copy number gains and losses [20,44,63,64,65], we integrated the miRNA and the array-CGH data from the same TNBC samples. This analysis, contrary to most of the studies performed in the literature, significantly reduced the technical variability in performing these molecular analyses in different samples and eliminated miRNA expression variations resulting from sample heterogeneity. As a result, a panel of 28 miRNAs was identified, with 16 miRNAs (miR-1204, miR-1224-5p, miR-1236-3p, miR-145-5p, miR-146a-5p, miR-2053, miR-218-5p, miR-3150b-3p, miR-3151-5p, miR-4448, miR-548d-3p, miR-548d-5p, miR-638, miR-661, miR-6721-5p, miR-765) presenting concordance with copy number alteration gains and/or losses. These miRNAs were most mapped to the 8q and 5q regions and were affected by copy number gains, and losses, respectively. Interestingly, pathway analysis demonstrated that 15 of the 16 miRNAs were situated in pathways associated with tumorigenesis, including the adherent’s junction and focal adhesion, proteoglycans in cancer, pathways in cancer, cAMP, Rap1, Ras, and pluripotency of stem cells signaling pathways. Additionally, the gene targets of these miRNAs included several cancer driver genes including *ZNF704* (targeted by ten miRNAs), *MMP16* (targeted by eight miRNAs), and *KCNN3, POU2F1, ADARB1, MPZL1, RUNX1T1, UBE2W,* and *DYRK1A* genes (targeted by seven miRNAs)**.** Some of these genes have been shown to confer aggressiveness to TNBC: *RUNX1T1*—identified to be associated with metastasis [66,67,68,69], *DYRK1B*—cell proliferation and mobility [70], *KCNN3*—cell proliferation, migration, and epithelial–mesenchymal transition, [71], and *ZNF704*—cell proliferation and poor prognosis [72]. Our analysis suggests that these genes may be commonly affected by both mechanisms of copy number and miRNA expression alterations. This supports the hypothesis that the mapping of miRNAs in regions with CNAs is not merely a physical finding but is biologically relevant.

Previously, we had identified a 17-miRNA signature in Brazilian Latina patients with TNBC using the integration of the copy number and miRNA expression. However, this signature was different from the one found in the current USA Latina study population. Nonetheless, several of the most significant signaling pathways were similarly affected, including the adherent’s junction, proteoglycans in cancer, pathways in cancer, and Rap1, Ras, and Hippo signaling pathways. It is noteworthy that the ancestral roots of the two Latina populations differed, with the Brazilian subjects tracing back to European origins whereas the USA subjects were from Central and South America.

To further our knowledge regarding miRNA expression levels of Latina TNBC, we compared the miRNA expression levels of the available TCGA data of TNBC and non-TNBC patients declared Latino and/or Hispanic, although there was no information on whether TCGA data were self-reported or ancestral genomically characterized. This comparison resulted in 99 differentially expressed miRNAs between the two groups, 23 of them common to the miRNAs differentially expressed in our TNBC and non-TNBC cases. Although these miRNAs were not within the panel of the 16-miRNA signature identified in our integration analysis, this observation may suggest an overall signature of miRNA expression common to TNBC of Latina patients.

It is relevant to highlight, as mentioned, that the differential miRNA expression among racial and ethnic groups can occur in individuals from the general population [22,25]. These expression differences are, however, mainly attributed to polymorphisms of SNPs that can occur in pre-miRNAs and mature miRNA binding sites and that exhibit varying allele frequencies in different populations [21,73]. These variants may influence miRNA expression; however, they do not necessarily impact cancer risk but contribute to population-specific miRNA expression differences [21,23] These small polymorphic alterations are, however, distinct from the somatic miRNA alterations of this study that were specifically detected in the TNBC tissue samples. In addition, the selection of miRNAs in this study, based on an integrated analysis with regions displaying large copy number alterations, were not observed in non-tumoral tissue and have been previously described in other cases of triple-negative breast cancer (TNBC). These approaches and evidence ensured the relevance and cancer specificity of the miRNAs identified in our samples as representative of the biology of the TNBC of the Latina patients studied.

Using bioinformatic analysis, we selected four miRNAs—miR-141-3p, miR-150-5p, miR-182-5p, and miR-661—to be individually validated in relation to their expression levels and tumor tissue specificity. We showed that these miRNAs present several target interactions to TNBC and/or breast cancer in general, and regulate cancer driver genes, such as *BRCA1*, *ESR1*, *PTEN*, and *AKT1.* MiR-141-3p and miR-182-5p were the miRNAs with the most central interactions, directly regulating a higher number of these gene targets.

The expression analysis revealed that miR-150-5p and miR-182-5p had the highest ability to discriminate tumor and non-tumor tissue (non-paired), with AUC values > 0.7. MiR-150-5p also presented the highest number of associations with the clinical parameters analyzed; its down-regulation was associated with larger tumor size, high expression levels of the p53 protein, increased breast cancer recurrence, presence of distant metastasis, and patients’ deceased status. Additionally, we found that the down-regulation of miR-150-5p was associated with a worse survival rate in the TCGA-BRCA patients, which suggests a tumor suppressive role for this miRNA in TNBC. However, our previous assays in TNBC indicated the opposite [74]. Overexpression of miR-150-5p was observed in tumor tissues compared with non-tumor tissues and in TNBC compared with non-TNBC tissues. High miR-150-5p levels were also associated with prolonged overall survival and increased cell proliferation, clonogenicity, migration, and drug resistance. For miR-182-5p, whose levels were observed up-regulated in the TNBC cases when compared to the non-tumor tissue, no association with the clinical–pathological parameters of the patients was found. In contrast to miR-150-5p expression, its down-regulation was associated with higher overall survival in the TCGA TNBC cases. Interestingly, a recent study [75] conducted in TNBC and non-TNBC cases of Brazilian patients also demonstrated up-regulation of miR-182-5p in TNBC cases compared to normal tissues. These data were also supported by another recent study [76] which demonstrated high expression of this miRNA in TNBC tissues and their corresponding plasma samples. In their TNBC cases, miR-182-5p up-regulation was associated with poor prognostic parameters, such as tumors with larger size, higher grades, and with tumor-infiltrated lymph nodes. Several other studies have evaluated the expression of miR-182-5p and its tumorigenic role in breast cancer, evidencing its relevance as a molecular biomarker with an oncogenic function [77,78,79,80,81]. MiR-141-3p and miR-661 were not significantly differentially expressed in the non-tumor tissues. However, up-regulation of miR-141-3p was associated with breast cancer recurrence, and down-regulation of miR-661 with larger tumor size. Increased levels of miR-141-3p were observed to inhibit the epithelial–mesenchymal transition of breast cancer cells [67]. Indeed, we demonstrated that the higher expression of miR-141-3p was associated with shortened survival in the TCGA TNBC patients’ analysis. MiR-661, which is located at the often highly amplified 8q23-24 chromosome region, is associated with basal tumors that present with focal amplification of the *C-MYC* oncogene [44,48]. This region has been noticeably amplified in TNBC, including the tumors containing *BRCA1* mutations [82,83], which are frequently in AA women [84]. Previously we reported up-regulation of miR-661 in TNBC of AA patients compared to non-TNBC [44]. We have also shown a differential expression of miR-661 between TNBC samples obtained from AA patients and Non-Hispanic White (NHW) women, which could indicate a potential association between miR-661 and race.

Collectively, our findings underscore the significant role of the identified miRNAs in influencing patient prognosis and clinical outcomes. Notably, among the four miRNAs validated, miR-141-3p, miR-150-5p, and miR-182-5p have been recognized as regulators of genes implicated in chemotherapy resistance and treatment response in breast cancer [79,85,86,87,88,89]. By conducting further investigations on TNBC Latina patients with comprehensive treatment information, along with long-term follow-up data, the specific contribution of these miRNAs to treatment response can be determined. Moreover, it can reveal novel therapeutic strategies that could be more effective in addressing the treatment needs of this population. Given the substantial impact of genetic heterogeneity observed in TNBC and its influence on treatment response, where miRNAs play an active role, integrating population-specific miRNA signatures that mediate treatment resistance become imperative for the success of therapeutic interventions. Such a precision medicine approach, tailored to the unique genetic makeup of TNBC among minority populations, such as Latinas, holds promise for improving treatment outcomes and reducing the breast cancer disparities in mortality rates of this population.

## 4. Materials and methods

### 4.1. Patients’ Accrual and Samples Collection

Twenty-eight formalin-fixed, paraffin-embedded (FFPE) TNBC tumor tissue sections were retrieved from the Pathology Center of the Hackensack University Medical Center (HUMC), New Jersey, USA. The cases were selected based on the initial patients’ self-reported information as Hispanics and/or Latinas. The samples were received in a coded fashion with no patient identifiers under the HUMC Institutional Review Board (IRB) approved protocol #1880. The TNBC subtype of the patients was determined by immunohistochemistry (IHC) analysis using ER, PR, and HER2 markers, following current guidelines [90,91].

Clinical–pathological information was obtained from the medical and pathology records deposited at the de-identified COTA, Inc. Real-World Data database and included age at diagnosis, tumor size and location, and expression of Ki67 and p53 proteins. Clinical follow-up information included breast cancer recurrence and distant metastasis, the presence of co-morbidities, and survival status. The mean age and tumor size of the patients were 55.3 ± 10.59 years and 1.85 ± 1.25 cm, respectively. Seven patients were diagnosed with bilateral breast cancer. Most patients (77.7%) presented high levels of Ki-67 (>10%) and were positive (77%) for p53 expression (>10%). The time of follow-up ranged from 28 to 76 months for the alive patients and 11–72 months for the deceased patients. Eight patients (8/26) presented with breast cancer recurrence and nine (9/28) developed distant metastasis, four of which to multiple sites. The most common metastatic sites were the lung, pleura, bone, and brain. Twenty (71.4%) patients presented co-morbidities; the most common was hypertension (40% of the patients), followed by diabetes, hypothyroidism, previous history of cancer (20%), myocardial diseases (15%), coronary disease and thrombosis (10%). The body mass index (BMI) mean value of the patients was 29.4 ± 6.7, with one patient presenting morbid obesity (BMI = 53.4). Most of the patients underwent neo-adjuvant therapy (ddACT regimen), followed by surgery and radiotherapy. Seven patients (6 of whom deceased) underwent multiple lines of therapy, including treatment with paclitaxel, carboplatin, capecitabine, gemcitabine, pembrolizumab, and eribulin.

### 4.2. Ancestral Markers Analysis

The patients of this study were initially selected from the COTA, Inc. database according to their self-reported ethnicity as Hispanics and/or Latinas. The ancestral information of 86% (24/28) of the patients was further confirmed by genotyping, using the SNP chip Illumina Infinium QC Array (Illumina Inc., San Diego, CA, USA), which contains about 3000 ancestral informative markers (AIMs), as previously described [20,39,44]. The genotype calling was performed using the Genome Studio software v. 2011.1. Genotypes from the mitochondrial genome and sex chromosomes were excluded, as well as genotypes with a call rate < 98%. The remaining autosomal genotypes (8687 in total) were integrated with the variant calling from ≥1900 individuals, originating from 21 diverse populations in the 1000 Genomes Project. To explore population structure among individuals, principal component analysis (PCA) was conducted on the genome-wide autosomal loci. First, a genetic relationship matrix was generated between pairs of individuals (GRM files) with the GCTA software [92]. Using the GRM files as input, the PCA method implemented in GCTA was applied using a default setting of 20 which outputted the first 20 eigenvectors and all the eigenvalues. Lastly, the top two principal components, PC1 and PC2, were plotted using RStudio (http://www.rstudio.com). This analysis showed that most of the patients clustered with or near the Latino populations, which, when refined, showed their clustering with individuals from Peru, Mexico, Colombia, and Puerto Rico, demonstrating the highly admixed background of the patients. Few patients clustered with the European- or African-derived populations (Figure 7).

### 4.3. General Study Design

A comprehensive integration of copy number and miRNA expression profiling was performed in the tumor tissue samples of ancestral genomic-characterized Latina patients with TNBC. Copy number and miRNA expression analyses were performed in the same tissue sections of the patients. The differentially expressed (DE) miRNAs (about controls-GEO database) that were mapped at the regions with copy number alterations (CNAs) were characterized by their main mRNA targets and their involvement in signaling pathways by functional enrichment and pathway analysis. A comparison of the DE miRNAs of the cases was conducted with the Hispanic/Latina TNBC and non-TNBC cases from the TCGA database. A subset of four miRNAs was validated by RT-qPCR in a subset of the tumor and adjacent non-tumor (ANT) tissue of the patients. The data were associated with the clinical–pathological, follow-up, treatment, and co-morbidities information of the patients (Figure 8).

### 4.4. Tissue Microdissection and DNA and RNA Isolation

A total of 10 µm formalin-fixed embedded (FFPE) unstained tissue sections were evaluated by the pathologist for the presence of at least 80% of the pure tumor cell population to ensure the absence of normal, necrotic, and/or inflammatory cells. The tumor cells were needle-micro dissected, and DNA and RNA were isolated as previously described [93]. DNA and RNA isolation was performed using phenol-chloroform, and Trizol (Invitrogen, Thermo Fisher Sci., Watham, MA, USA), respectively. The concentration and purity of DNA and RNA were assessed by the NanoDrop Spectrophotometer (Thermo Fisher Sci.).

### 4.5. Array Comparative Genomic Hybridization (Array-CGH) and Analysis

DNA copy number alterations (CNAs) were evaluated by array-CGH analysis using the Agilent platform (Agilent Tech., Santa Clara, CA, USA) as previously described [93]. Briefly, equal amounts of the isolated tumor and reference genomic DNA (a pool obtained from multiple female individuals with no cancer) (100–300 ng) were digested and labeled using a SureTag Complete DNA Labeling Kit (Agilent Tech.) and hybridized in the arrays for 40 h. Only cases that showed satisfactory incorporation of more than 1.00 pico/mol labeling were selected for hybridization. The array data were extracted using Feature Extraction (FE) software v10.10, and the Agilent Cytogenomic v. 7.0 software (Agilent Tech.) was used to analyze the data using the aberration detection method-ADM2, a threshold of 6.0, and defined aberration filters. Copy number gains and losses were considered when present in at least 3 consecutive probes with values of mean absolute log2 ratio (intensity of the Cy5 dye (reference DNA)/intensity of the Cy3 dye (test DNA) value of ≥0.25 and ≤−0.25, respectively) as per our previous analysis [20]. UCSC Genome Browser (GRCh37/hg19) and miRbase 22.1 databases were used to determine the genes and miRNAs present in each selected cytoband affected by CNA, respectively.

### 4.6. Global miRNA Expression Analysis and Statistical Analyses

MiRNA expression profiling was performed using the NanoString nCounter technology Human v3 miRNA Expression Assay (Seattle, WA, USA) according to our previous protocols [20,44]. This miRNA panel contains 827 endogenous miRNA human probes derived from miRbase v.18, 6 negative controls, 6 positive controls, 3 ligation positive controls, 3 ligation negative controls, 5 spike-in controls, and 5 housekeeping transcripts (*B2M, ACTB, GAPDH, RPL19,* and *RPLP0*). The raw miRNA expression data were pre-processed using NanoString’s nCounter RCC collector worksheet. As a control group for the TNBC subtype specificity, the miRNA expression data of [94] was used, composed of 32 (18 ER−/PR+ and 14 ER+/PR−) single-hormone positive breast tumor samples. The RCC files were downloaded from Gene Expression Omnibus (GEO) with the accession number GSE155362. Each RCC file (TNBC and control groups) was uploaded, the background was subtracted (negative control geometric mean), and the data were normalized (positive control normalization: geometric mean; and CodeSet Content normalization to all genes, geometric mean) using NanoString’s nSolver 4.0 software. Unsupervised (UHC) and supervised hierarchical cluster (SHC) analysis were performed on significantly differentially expressed miRNAs among the patients’ subtypes, using Pearson’s correlation coefficient, average linkage, and Benjamini–Hochberg multiple testing correction on the Multiexperiment Viewer software (MeV 4.9) (*t*-test *p* < 0.01, FDR < 0.05).

### 4.7. Integrated Analysis of Array-CGH and miRNA Data

Integration of the most DE miRNAs associated with the TNBC subtype with array-CGH data from the same samples was performed using two distinct approaches, as previously described [44,95]. Briefly, the first approach consisted of the mapping of the miRNAs at the cytobands most affected by CNAs and further selection was based on their concordance level (i.e., cytobands with copy number gains/amplifications/up-regulated miRNA expression and cytobands with copy number losses/deletions/down-regulated miRNA expression). The location of each miRNA was determined using miRBase (http://www.mirbase.org) v.22.1. The second approach was based on the identification of common genes that are targets of the above-selected miRNAs and may be affected by both CNAs and miRNA expression alterations. A list of the predicted target genes for each miRNA was constructed using the online available databases: Diana micro-T-CDS v.5.0 (http://diana.imis.athena-innovation.gr/DianaTools/index.php?r=MicroT_CDS/index), miRDB (http://www.mirdb.org/miRDB/, and TargetScan Release 8.0 (http://www.targetscan.org/vert_71/). Only miRNA target genes that were present in two out of the three miRNA databases were selected.

### 4.8. Biological Function and Pathway Analysis

To assess the potential impact of the deregulated above-identified miRNAs in the TNBC biological processes and pathways, Diana miRPath v.3.0 was used (http://diana.cslab.ece.ntua.gr). Enrichment analysis of multiple miRNA target genes comparing each set of miRNA targets to all known KEGG (Kyoto Encyclopedia of Genes and Genomes) pathways [96] was obtained and selected by significant *p*-value (*p* < 0.05) and cancer-associated biological functions.

### 4.9. The Cancer Genome Atlas (TCGA) Data Processing and Analysis

Total RNA-seq data from 33 breast cancer cases (7 TNBC and 26 non-TNBC) classified as Hispanic/Latina women were obtained from The Cancer Genome Atlas (TCGA) using the GDCRNATools R package [97]. Differential expression (DE) analysis was performed comparing the TNBC with the non-TNBC samples using the GDCRNATools package applying the Limma method [98], considering only miRNAs with logFC >1.5 and *p*-value ≤ 0.01.

### 4.10. Selection of miRNA for RT-qPCR Expression Analysis and miRNA–mRNA Network Construction

The Integrated Breast Cancer Pathway (Wikipathways), which consists of the integration of critical proteins involved in breast cancer based on the Human Pathway Database (HPD) [99], was used to select genes involved in breast cancer pathogenesis. Two miRNAs experimentally validated targets databases were used to identify miRNAs potentially involved in this pathway: miRTarbase v.9 (based on strong and weak validation evidence) and Diana Tarbase v.8 (based on low and high throughput experiments) [100,101]. The identified subset of targets was further analyzed using network analysis based on the STRING (v.11) [102] protein–protein interaction database (http://string-db.org) and the Cytoscape software v. 3.9.1 [103] as used to construct a miRNA–mRNA network with selected miRNAs and their respective target genes, as previously performed [74].

### 4.11. Quantitative Reverse Transcription Polymerase Chain Reaction (RT-qPCR) Analysis

The previously isolated RNA from 18 out of the 28 TNBC tumor and adjacent non-tumor (ANT) FFPE tissue sections were subject to RT-qPCR using TaqMan miRNA assays (Applied Biosystems, Thermo Fischer Sci) with TaqMan probes for miR-141-3p (assay #000463), miR-150-5p (assay #000473), miR-182-5p (assay #002334), and miR-661 (assay #001606), as previously described [74]. Tissue samples were normalized to RNU48. Samples with threshold cycle (Ct) values of ≥31 for RNU48 and ≥35 for the other miRNAs were excluded from the analysis. Each reaction was performed in triplicate, and the mean value of the three-cycle threshold was used and presented as means ± SE, considering the *p*-value ≤ 0.05.

### 4.12. Receiver Operating Characteristic (ROC) Curve Analysis

ROC analysis to calculate the area under the curve (AUC) was performed by GraphPad Prism 8.0.2 (GraphPad Software Inc., San Diego, CA, USA) to identify the discriminatory power of the four selected miRNAs in differentiating the TNBC tumor and the ANT tissues. Sensitivity was plotted against 1-specificity for the binary classifier (TNBC and ANT). An AUC of 100% denotes perfect discrimination by the miRNA, whereas an AUC of 50% denotes a complete lack of discrimination by the miRNA. AUCs and 95% corresponding confidence intervals were calculated for each miRNA and the combined miRNAs.

### 4.13. Association of the RT-qPCR Results with the Patients’ Clinical–Pathological Data

The association of miRNA expression levels obtained by RT-qPCR with the patients’ clinical–pathological parameters, comorbidities, BMI values, and follow-up data were performed using the GraphPad Prism 8.0.2 with an unpaired *t*-test and Welch’s correction. A multivariate linear regression analysis was conducted to evaluate the relationship between the miRNAs and the clinical–pathological parameters and other information described above, assuming no correlation or covariance exist amongst the miRNAs. *p* ≤ 0.05 was considered significant.

### 4.14. Survival Analysis

Survival analysis of the patients was performed using GraphPad Prism 8.0.2. For both TNBC and non-TNBC cases, *p* ≤ 0.05 was considered significant, using the Log-rank test for trend. In addition, the Kaplan–Meier (KM) plots (https://kmplot.com/) of data from the METABRIC and TCGA databases were constructed based on the expression of miR-141-3p, miR-150-5p, miR-182-5p, and miR-661 in breast cancer samples of all molecular subtypes and TNBC only.

## 5. Conclusions

This study highlights the importance of considering patient ancestry in breast cancer as it can influence the relationship of tumor molecular signatures and clinical manifestations of the disease. Despite its clinical potential, information is scarce regarding miRNA signatures in Latinas/Hispanic breast cancer populations. Herein, we have identified a miRNA signature of TNBC among ancestral genomic-characterized Latina patients using integration with copy number analysis. This integration identified miRNAs that are potentially regulated by either gains and/or losses and miRNAs that may be involved in the regulation of target genes. These findings provide a more comprehensive understanding of the interplay between molecular mechanisms in cancer. Mechanistic analyses of the interactions of miRNAs and their gene targets identified are required to discern the role of these integrated molecular signatures in TNBC. In addition, the miRNAs observed differentially expressed in the TNBC of the Latina patients of our study can potentially interact with non-biologic factors to promote unique tumor characteristics. The relative contributions of these biologic and non-biologic factors need to be investigated in larger studies of TNBC of Latina patients to determine their interaction and clinical impact.

## Figures and Tables

**Figure 1 ijms-24-13046-f001:**
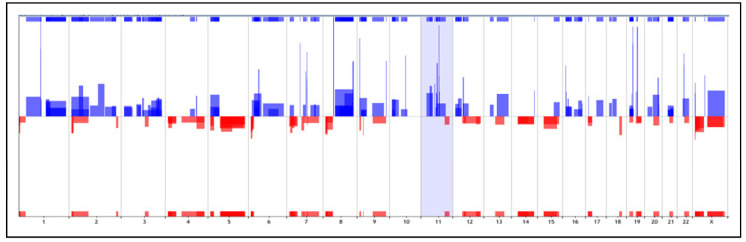
Genomic view/penetrance plot of array-CGH profiling among analyzed TNBC cases (n = 24). Vertical lines represent the number of each chromosome. Blue peaks indicate copy number gains, and red peaks indicate copy number losses (as highlighted for chromosome 11). (Agilent Genomic Workbench 7.0.)

**Figure 2 ijms-24-13046-f002:**
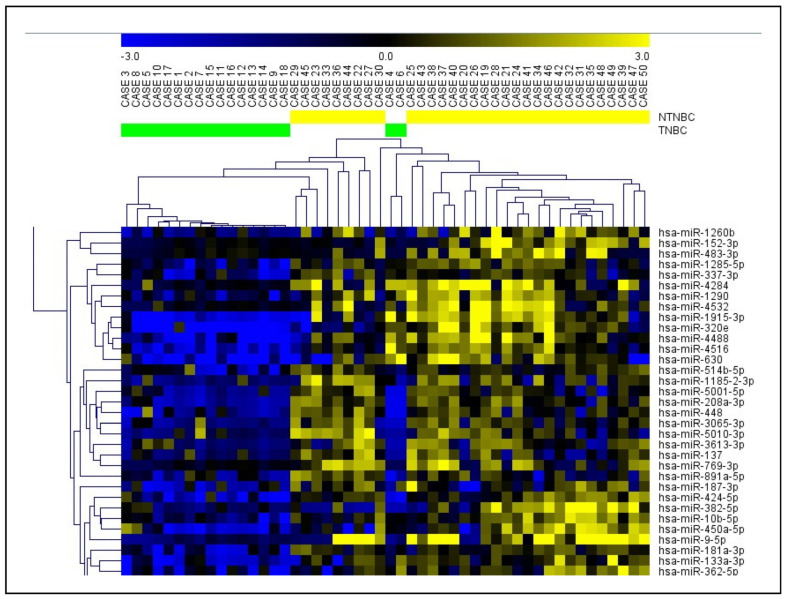
Supervised hierarchical clustering (SHC) analysis of global miRNA expression in TNBC and non-TNBC cases (MeV 4.9, *t*-test *p* < 0.01, FDR < 0.05). The selected area of the heatmap is shown.

**Figure 3 ijms-24-13046-f003:**
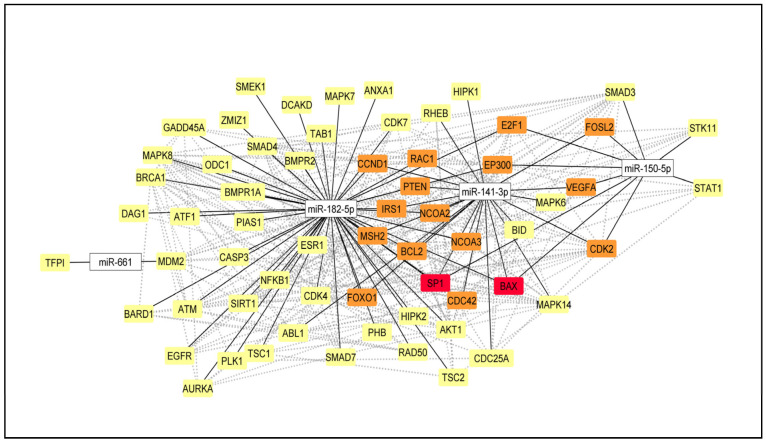
miRNA–mRNA network based on the target genes of miR-141-3p, miR-150-5p, miR-182-5p, and miR-661 (obtained from the Integrated Breast Cancer Pathway (Wikipathways)). mRNAs highlighted with yellow color targeted by one miRNA, orange color targeted by two miRNAs, and red color targeted by all four miRNAs. Solid and dashed lines represent protein–protein and miRNA–mRNA interactions, respectively (network edited by Cytoscape v.3.0).

**Figure 4 ijms-24-13046-f004:**
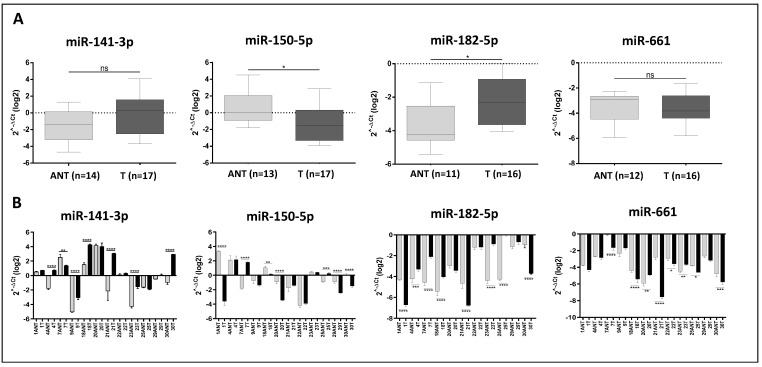
RT-qPCR miRNA expression analysis of miR-141-3p, miR-150-5p, miR-182-5p, and miR-661 in adjacent non-tumor (ANT) tissue and tumor tissue groups of the TNBC patients (**A**) and in the paired cases (**B**). Unpaired *t*-test. * *p* ≤ 0.05, ** *p* ≤ 0.01, *** *p* ≤ 0.001, **** *p* ≤ 0.0001, ns, not significant.

**Figure 6 ijms-24-13046-f006:**
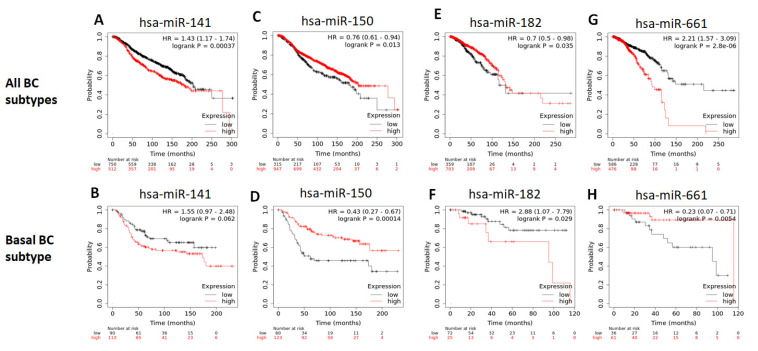
Expression of miR-141-3p, miR-150-5p, miR-182-5p, and miR-661 and their associations with the survival of breast cancer patients of the METABRIC(**A**–**D**) and TCGA datasets (**E**–**H**). Upper panel, breast cancer cases of all molecular subtypes; lower panel, breast cancer cases of the basal subtype.

**Figure 7 ijms-24-13046-f007:**
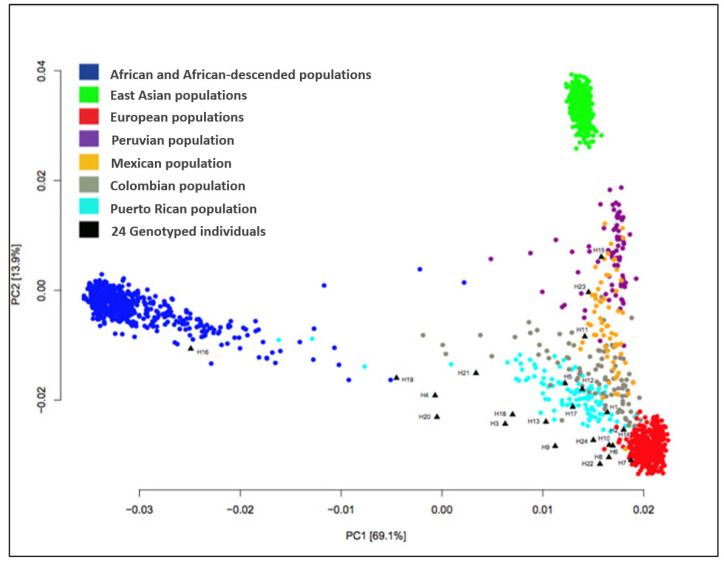
PCA plot of the ancestral marker analysis of this study’s Latina patients (black triangles) merged with the 1000 Genomes Project reference populations (R studio).

**Figure 8 ijms-24-13046-f008:**
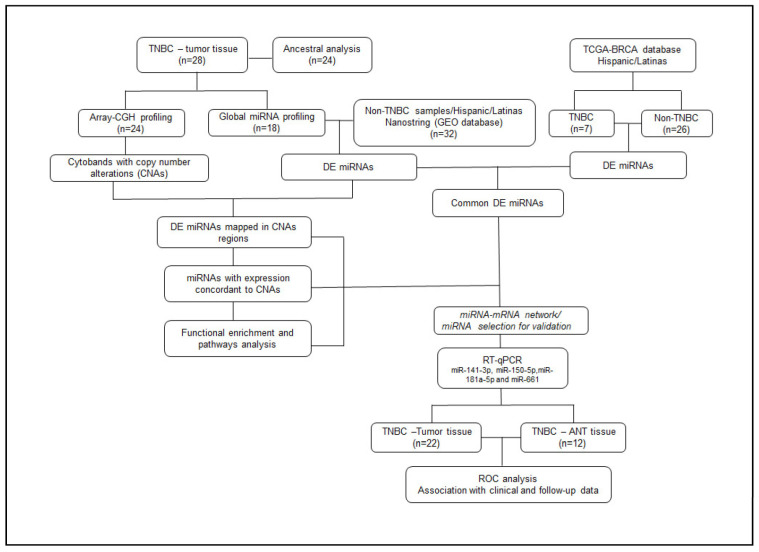
Workflow of the study general design.

**Table 1 ijms-24-13046-t001:** The most frequent cytobands and corresponding genomic annotations affected by CNAs identified by array-CGH analysis of the studied Latina patients with TNBC.

Chr	Cytoband	Start	Stop	Size (kb)	Cases (%)	Gains/Losses	of Probes
chr1	q21.1–q24.2	144,374,546	1.68 × 10^8^	23,435,182	6 (28.5%)	Gain	533
chr3	q26.1–q27.2	166,346,288	1.85 × 10^8^	18,991,985	5 (23.8%)	Gain	132
chr4	p16.3–p15.31	1,914,109	20,323,997	18,409,889	4 (19.01%)	Loss	243
chr5	q21.1–q35.3	99,381,621	1.72 × 10^8^	72,831,986	4 (19.01%)	Loss	1048
chr6	p25.3–p24.2	248,239	10,815,671	10,567,433	8 (33.33%)	Loss	16
chr6	p22.3–p21.32	17,745,590	32,262,768	14,517,179	4 (19.01%)	Gain	321
chr8	q13–q24.3	69,999,338	1.46 × 10^8^	75,139,299	7 (33.33%)	Gain	1038
chr8	q24.3	141,355,101	1.45 × 10^8^	3,898,583	7 (33.33%)	Gain	587
chr11	q13.2–q13.3	68,249,411	70,012,823	1,763,413	4 (19.01%)	Gain	31
chr19	p13.3–p13.11	781,586	17,833,369	17,051,784	7 (33.33%)	Gain	560
chr21	q21.3–q22.3	28,834,275	45,382,723	16,549,449	4 (19.01%)	Gain	344
chrX	p22.33	1,179,089	2,353,577	1,174,489	7 (33.33%)	Loss	42
chrX	p22.33	218,292	2,622,294	2,404,303	5 (23.8%)	Gain	75
chrX	p22.33–p22.2	2,662,039	13,621,701	10,959,663	10 (47.6%)	Loss	149

Chr: chromosome, amp: amplification; del: deletion.

**Table 2 ijms-24-13046-t002:** Top differentially expressed 15 miRNAs between the TNBC non-TNBC cases are presented by fold change (FC) value.

Up-Regulated			Down-Regulated		
miRNA	FC (log2)	*p*-Value	MiRNA	FC (log2)	*p*-Value
hsa-miR-661	4.12434	1.75 × 10^−13^	hsa-miR-141-3p	−4.11176	1.00 × 10^−6^
hsa-miR-1270	4.04835	<0.0001	hsa-miR-125a-5p	−5.17837	2.40 × 10^−8^
hsa-miR-548	3.92366	<0.0001	hsa-miR-222-3p	−5.24417	1.10 × 10^−8^
hsa-miR-548h-3p	3.92366	<0.0001	hsa-let-7b-5p	−5.27658	1.00 × 10^−7^
hsa-miR-517a-3p	3.72959	1.11 × 10^−15^	hsa-miR-29b-3p	−5.32329	5.97 × 10^−7^
hsa-miR-548al	3.66887	1.33 × 10^−8^	hsa-miR-15a-5p	−5.33016	1.86 × 10^−7^
hsa-miR-765	3.65976	4.88 × 10^−15^	hsa-miR-200c-3p	−5.35346	1.52× 10^−6^
hsa-miR-761	3.58448	5.15 × 10^−11^	hsa-miR-4286	−5.44997	1.26× 10^−6^
hsa-miR-219b-3p	3.38808	7.44 × 10^−9^	hsa-miR-93-5p	−5.46019	2.19 × 10^−7^
hsa-miR-605-5p	3.14118	4.53 × 10^−8^	hsa-miR-126-3p	−5.70673	3.22 × 10^−7^
hsa-miR-608	3.11135	4.31 × 10^−10^	hsa-miR-181a-5p	−5.71558	4.93 × 10^−7^
hsa-miR-212-3p	3.04930	1.88 × 10^−13^	hsa-miR-21-5p	−5.98850	1.52× 10^−6^
hsa-miR-508-3p	3.00980	7.44 × 10^−10^	hsa-miR-191-5p	−6.27011	4.31 × 10^−8^
hsa-miR-219a-5p	3.00575	3.08 × 10^−8^	hsa-miR-150-5p	−6.33220	7.32 × 10^−8^
hsa-miR-325	2.95349	9.22 × 10^−10^	hsa-let-7a-5p	−8.81173	5.50 × 10^−7^

**Table 3 ijms-24-13046-t003:** Differentially expressed miRNAs between the TNBC and non-TNBC cases with expression levels in concordance with copy number alterations (CNAs) are presented by fold change (FC) value.

miRNA	FC (log2)	*p*-Value	FDR	Cytoband	CNA
hsa-miR-661	4.12434	1.75 × 10^−13^	1.75 × 10^−11^	8q24.3	gain
hsa-miR-765	3.65976	4.88 × 10^−15^	9.75 × 10^−13^	1q23.1	gain
hsa-miR-3151-5p	2.63406	7.04 × 10^−7^	6.46 × 10^−6^	8q22.3	gain
hsa-miR-2053	2.94650	1.75 × 10^−8^	3.49 × 10^−7^	8q23.3	gain
hsa-miR-548d-5p	1.98169	6.27 × 10^−6^	3.58 × 10^−5^	8q24.13	gain
hsa-miR-6721-5p	1.58740	4.62 × 10^−4^	0.00129	6p21.32	gain
hsa-miR-548d-3p	1.50149	2.90 × 10^−5^	1.32 × 10^−4^	8q24.13	gain
hsa-miR-638	1.17150	1.68 × 10^−4^	5.82 × 10^−4^	19p13.2	gain
hsa-miR-1224-5p	0.99921	3.44 × 10^−4^	0.00103	3q27.1	gain
hsa-miR-3150b-3p	0.49202	2.65 × 10^−4^	8.55 × 10^−4^	8q22.1	gain
hsa-miR-1204	0.40531	6.10 × 10^−4^	0.00150	8q24.21	gain
hsa-miR-4448	0.39551	7.85 × 10^−4^	0.00179	3q27.1	gain
hsa-miR-218-5p	−2.71112	5.71 × 10^−8^	9.12 × 10^−7^	5q34	loss
hsa-miR-146a-5p	−2.81117	1.85 × 10^−6^	1.35 × 10^−5^	5q33.3	loss
hsa-miR-145-5p	−4.69634	8.57 × 10^−8^	1.24 × 10^−6^	5q32	loss

**Table 4 ijms-24-13046-t004:** Common differentially expressed miRNAs between the TNBC and non-TNBC cases of the TCGA breast cancer Latina cases and the cases of this study.

	TNBC vs. Non-TNBC (This Study)			TNBC vs. Non-TNBC TCGA		
	FC (log2)	*p*-Value	FDR	FC (log2)	*p*-Value	FDR
hsa-let-7a-5p	−8.81174	5.50 × 10^−7^	5.29 × 10^−6^	−1.03695	0.004	0.06594
hsa-let-7b-5p	−5.27658	1.00 × 10^−7^	1.40 × 10^−6^	−1.15425	0.010	0.10343
hsa-let-7f-5p	−4.62780	2.86 × 10^−7^	3.26 × 10^−6^	−1.05391	0.006	0.07720
hsa-let-7g-5p	−4.45426	2.27 × 10^−6^	1.59 × 10^−5^	−0.57203	0.032	0.17183
hsa-miR-10a-5p	−2.32238	5.63 × 10^−9^	1.40 × 10^−7^	−1.60710	0.005	0.07131
hsa-miR-10b-5p	−1.86443	4.05 × 10^−6^	2.55 × 10^−5^	−0.93191	0.022	0.16072
hsa-miR-146b-3p	0.40531	6.10 × 10^−4^	0.001503	1.46182	0.002	0.03960
hsa-miR-181c-5p	−1.87328	8.15 × 10^−5^	3.24 × 10^−4^	−0.82176	0.045	0.21220
hsa-miR-191-5p	−6.27011	4.31 × 10^−8^	7.16 × 10^−7^	−1.33551	0.000	0.01904
hsa-miR-195-5p	−2.90058	4.43 × 10^−7^	4.53 × 10^−6^	−1.10876	0.044	0.20910
hsa-miR-200a-3p	−2.42572	1.90 × 10^−4^	6.41 × 10^−4^	−1.31242	0.011	0.11347
hsa-miR-200b-3p	−5.03990	2.95 × 10^−6^	1.93 × 10^−5^	−1.07680	0.016	0.14109
hsa-miR-26a-5p	−4.78689	2.76 × 10^−7^	3.24 × 10^−6^	−0.70270	0.042	0.20410
hsa-miR-26b-5p	−4.82779	1.57 × 10^−7^	2.03 × 10^−6^	−0.95776	0.017	0.14177
hsa-miR-29b-3p	−5.32329	5.97 × 10^−7^	5.68 × 10^−6^	−0.95562	0.031	0.16993
hsa-miR-29c-3p	−4.26978	2.44 × 10^−8^	4.64 × 10^−7^	−1.67903	0.000	0.01904
hsa-miR-30a-5p	−3.51938	1.90 × 10^−5^	9.15 × 10^−5^	−1.90155	5.71 × 10^−5^	0.00646
hsa-miR-30b-5p	−4.62253	8.25 × 10^−8^	1.24 × 10^−6^	−0.89401	0.003	0.05813
hsa-miR-342-3p	−4.73665	8.75 × 10^−9^	2.05 × 10^−7^	−1.49716	0.005	0.06655
hsa-miR-34a-5p	−2.44624	3.34 × 10^−7^	3.65 × 10^−6^	−0.72461	0.025	0.16814
hsa-miR-423-5p	−2.72085	7.42 × 10^−6^	4.14 × 10^−5^	−0.84124	0.021	0.16072
hsa-miR-664a-3p	−1.71081	1.43 × 10^−4^	5.20 × 10^−4^	−0.87318	0.021	0.16072
hsa-miR-766-3p	2.89894	1.71 × 10^−9^	5.93 × 10^−8^	1.20479	0.000	0.02093

**Table 5 ijms-24-13046-t005:** Association of the expression levels of miR-141-3p, miR-150-5p, miR-182-5p, and miR-661 with the patient’s clinical characteristics.

Clinical Variable	miR-141-3p	miR-150-5p	miR-182-5p	miR-661
Age at diagnosis	n = 17, *p* = 0.753	n = 17, *p* = 0.410	n = 16, *p* = 0.646	n = 17, *p* = 0.111
>55.5, ≤55.5				
Tumor size (cm)	n = 17, *p* = 0.218	n = 16, ***p* = 0.01**	n = 16, *p* = 0.537	n = 17, ***p* = 0.013**
>1.7, ≤1.7				
Tumor grade	n = 15, *p* = 0.434	n = 15, *p* = 0.803	n = 15, *p* = 0.273	n = 15, *p* = 0.198
2, 3				
Tumor stage	n = 17, *p* = 0.968	n = 17, *p* = 0.569	n = 16, *p* = 0.597	n = 17, *p* = 0.704
T1/T2, T3/T4				
Ki-67	n = 16, *p* = 0.272	n = 16, *p* = 0.630	n = 15, *p* = 0.236	n = 16, *p* = 0.969
>10%, ≤10%				
p53	n = 16, *p* = 0.138	n = 16, ***p* = 0.001**	n = 15, *p* = 0.297	n = 16, *p* = 0.406
>10%, ≤10%				
BC recurrence	n = 16, ***p* = 0.007**	n = 16, ***p* = 0.009**	n = 16, *p* = 0.489	n = 17, *p* = 0.510
Yes/No				
Dist Mets	n = 17, *p* = 0.437	n = 16, ***p* = 0.011**	n = 16, *p* = 0.321	n = 17, *p* = 0.889
Yes/No				
Survival status	n = 17, *p* = 0.846	n = 17, ***p* = 0.021**	n = 16, *p* = 0.459	n = 17, *p* = 0.654
Alive/Deceased				
BMI values	n = 17, *p* = 0.429	n = 17, *p* = 0.132	n = 16, *p* = 0.627	n = 17, *p* = 0.157
>28.3, ≤28.3				
Co-morbidities	n = 17, *p* = 0.155	n = 17, *p* = 0.766	n = 16, *p* = 0.982	n = 17, *p* = 0.779
Yes/No				
HTN	n = 17, *p* = 0.173	n = 17, *p* = 0.479	n = 16, *p* = 0.482	n = 17, *p* = 0.160
Yes/No				

BC: breast cancer; Dist Mets: distant metastasis, BMI: body mass index; HTN; hypertension.

## Data Availability

All the data is provided in the manuscript text and in the Appendix A.

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
