# Peer review of "Deregulated miRNA Expression in Triple-Negative Breast Cancer of Ancestral Genomic-Characterized Latina Patients"

_ijms, 2023, doi:10.3390/ijms241713046_

Round 1
Reviewer 1 Report
The article by Almohaywi et al. investigates the deregulated microRNA (miRNA) expression in triple-negative breast cancer (TNBC) in Latina patients. The study profiles TNBC biopsies for genome-wide copy number and miRNA expression and identifies a panel of 28 miRNAs associated with copy number alterations (CNAs). The study validates four selected miRNAs and discusses their association with various clinical features. The study emphasizes the importance of considering ancestral background in examining TNBC.
General Comments:
Clarity and Methodology:
The language used in the article is clear, and the methods employed are adequately described. The study design is coherent, and the data analysis seems to be robust.
Research Question:
The introduction of the article should provide a clearer explanation of the research question and its significance. It is not evident why the authors chose to investigate this particular question. The rationale behind focusing on CNAs and miRNA expression, as opposed to Whole Exome Sequencing (WES) or RNA-Seq, should be clarified. The authors should elucidate how this study can advance medicine or contribute to the existing body of knowledge.
Multivariate Cox Regression Analysis:
The authors identify a set of miRNAs that are deregulated in their cohort. It is recommended that the authors perform a multivariate Cox regression analysis to evaluate the association between clinical variables and miRNA expression in relation to overall survival (OS) and other outcomes in these patients. This analysis would strengthen the conclusions and provide more insights into the clinical significance of the findings.
Role of miRNA in Treatment Response:
The article does not seem to discuss the role of miRNA expression in response to treatment. It is recommended that the authors include a section in the discussion that addresses this aspect. Understanding the role of miRNA in treatment response could have implications for personalized medicine and treatment strategies for TNBC in Latina patients.
Prediction of miRNA Target Genes:
The authors should consider running prediction algorithms to identify genes that are targeted by the miRNAs identified in this study. Incorporating these target genes into survival models and validating them in TCGA datasets, as the authors have done in Figure 6, would add value to the study. This could provide insights into the molecular mechanisms underlying the associations observed.
Reviewer 2 Report
In this paper, authors studied miRNA as biomarkers for breast cancer in women with Hispanic and Latin American ancestry. They mainly focused on 4 miRNAs, and revealed their up and/or down regulation profile associate with triple-negative breast cancer progression and prognosis. Overall, miRNA expression and their potential as cancer biomarker is an interesting topic to general audience. However, this particular study only recruited 28 patients and most of the data sets have less than 20 data points, thus, it is hard to judge whether those 28 New Jersey patients could represent the broad Hispanic and/or Latin American population. In addition, a very similar study was published about Brazil (Latin American) population in 2019 (Oncotarget. 2019 Oct 22; 10(58): 6184–6203.) with very similar methods and analysis applied. Thus, identifying a few more miRNAs from a limited population (28 patients) that may or may not lead to clinical outcome is hard to justify the novelty and significance of this paper.
Some detailed comments please see below.
Detailed comments:
1. Can the authors comment on the major differences or improvements this paper provides compared to the previous published studies on similar topic?
2. Could the authors please comment on the miRNA biomarkers in triple-negative breast cancer patients with different ancestry than Latin America? How does their miRNA biomarker different from the Latin America population’s biomarker that led the authors conclude ancestry is an important factor in clinical manifestations of the disease?
3. What about the miRNA expression level in healthy adults with different ancestry? Is it possible that the miRNA differences are more associated with ancestry regardless of the disease state?
4. What are the health status of the patients despite of breast cancer? Is it possible that miRNA level is also related to other disease complications?
5. Is there a mechanism of why some miRNA is important for certain ancestry?
6. What really missing from this study is the
7. Page 2 line 58-59 incomplete sentence.
8. Page 2 line 70, I would move TNBC explanation to where it first appear in the paragraph above
9. Page 7 line 199-200 incomplete sentence
Round 2
Reviewer 1 Report
The authors addressed carefully all the comments raised upon the peer review process. The manuscript is now suitable for acceptance.
Author Response
Thank you.
Reviewer 2 Report
The authors successfully addressed all the questions/comments I have. No additional questions araise. Only one minor suggestion: discussions from comment #1 (significance of the study), 2, 4 (important controls), and 5 (mechanistic discussion and learnings moving forward) are all very great points that might worth incorporating into the manuscript.
Author Response
As per the reviewer's recommendation, we have incorporated into the manuscript the discussions related to the comments previously raised.
Thank you!
